# Thiophene Derivatives as Anticancer Agents and Their Delivery to Tumor Cells Using Albumin Nanoparticles

**DOI:** 10.3390/molecules24010192

**Published:** 2019-01-06

**Authors:** Guangsheng Cai, Simiao Wang, Lang Zhao, Yating Sun, Dongsheng Yang, Robert J. Lee, Menghui Zhao, Huan Zhang, Yulin Zhou

**Affiliations:** 1College of Life Sciences, Jilin University, Changchun 130012, China; caigs15@mails.jlu.edu.cn (G.C.); wangsm1317@mails.jlu.edu.cn (S.W.); 18744026372@sina.cn (Y.S.); lee.1339@osu.edu (R.J.L.); zhaomh18@mails.jlu.edu.cn (M.Z.); 2College of Chemistry, Jilin University, Changchun 130012, China; zlang2009@163.com; 3School of Pharmaceutical and Food Sciences, Zhuhai College of Jilin University, Zhuhai 519041, China; 07034@jluzh.com; 4Division of Pharmaceutics and Pharmaceutical Chemistry, College of Pharmacy, The Ohio State University, Columbus, OH 43210, USA

**Keywords:** thiophene derivatives, human serum albumin, drug delivery, nanoparticles, cytotoxicity

## Abstract

A series of thiophene derivatives (TPs) were synthesized and evaluated for cytotoxicity in HepG2 and SMMC-7721 cell lines by MTT assay. TP **5** was identified as a potential anticancer agent based on its ability to inhibit tumor cell growth. Drawbacks of TPs, including poor solubility and high toxicity, were overcome through delivery using self-assembling HSA nanoparticles (NPs). The optimum conditions for TP **5**-NPs synthesis obtained by adjusting the temperature and concentration of TP **5**. The NPs had an encapsulation efficiency of 99.59% and drug-loading capacity of 3.70%. TP **5** was slowly released from TP **5**-NPs in vitro over 120 h. HepG2 and SMMC-7721 cell lines were employed to study cytotoxicity of TP **5**-NPs, which exhibited high potency. ROS levels were elevated and mitochondrial membrane potentials reversed when the two cell lines were treated with TP **5**-NPs for 12 h. Cellular uptake of fluorescence-labeled TP **5**-NPs in vitro was analyzed by flow cytometry and laser confocal scanning microscopy. Fluorescence intensity increased over time, suggesting that TP **5**-NPs were efficiently taken up by tumor cells. In conclusion, TP **5**-NPs showed great promise as an anticancer therapeutic agent.

## 1. Introduction

Cancer is a leading cause of death and its rate has been increasing [1,2]. Heterocyclic compounds, including thiophene derivatives, indole, and thiazolone, have attracted considerable attention due to their antitumor activities [3,4,5]. In our previous works, we have synthesized a series of thiophene derivatives (TPs) [3], which were shown to have significant anti-tumor activities [5,6]. However, low solubility and high toxicity have limited their utility [7]. Therefore, it is desirable to develop a drug delivery system that can enhance their solubility, tumor bioavailability and reduce the side effects of TPs.

Nanovehicles, including polymers, lipid vesicles, dendrimers, and polymer-proteins, have been shown to improve the solubility of hydrophobic drugs [8,9]. Moreover, they have been shown to passively accumulate in tumor tissue via enhanced permeability and retention (EPR) effect [10,11,12,13]. Currently, with the approval of Abraxane^®^ (albumin-paclitaxel nanoparticles) by the FDA, human serum albumin (HSA)-based nanoparticles have attracted much attention [14]. HSA is a major component of plasma and has advantages of being non-toxic, non-immunogenic, biodegradable, biocompatible, and easy to modify and dissolve in water [15,16,17]. Meanwhile, hydrophobic drugs are easy to incorporate into the particle matrix because of a multitude of drug binding sites presented in the albumin molecule. In addition, extracellular matrix glycoprotein (Secreted Protein, Acidic and Rich in Cysteine–SPARC), which has a high affinity for albumin and enriched in tumors, can further enhance its tumor deposition [18]. Therefore, TPs-loaded HSA-based nanoparticles could enhance its solubility, tumor bioavailability and reduce their side effect.

In this work, 6 TPs were synthesized and evaluated for cytotoxicity in HepG2 and SMMC-7721 cell lines. In order to improve the solubility of TPs, TPs albumin nanoparticles (TPs-NPs) were prepared by a simple method with self-assembling HSA nanoparticles, which is presented in Scheme 1. Temperature and the concentration of TPs were discussed to determine the optimum parameters for preparing TPs-NPs. Transmission electron microscopy (TEM) was used to observe morphology and microstructure of TPs-NPs. Properties of TPs-NPs, including particle size, zeta potential, stability, drug loading efficiency, and drug release were investigated. Besides, cellular uptake of TPs 5-NPs in vitro was quantitative and qualitative analyzed by flow cytometry and laser confocal scanning microscopy (CLSM). In addition, cellular apoptosis was investigated by mitochondrial transmembrane potential and reactive oxygen species (ROS) assay.

## 2. Results

### 2.1. MTT Screening Assay

Six TPs possessing 2,3-fused thiophene scaffolds were synthesized and their structures are presented in Figure 1A, which have been reported previously [3]. The cytotoxicity was investigated in HepG2 and SMMC-7721 cell lines by MTT screening assay, and the data are presented in Figure 1B. As shown in Figure 1B, for two cell lines, all TPs showed certain antitumor activity and cell viability was all 18–98.0% when the concentration of TPs was 30.0 μg/mL, suggesting TPs can induce tumor cell death. However, by comparison, TP **5** displays higher activity than other TPs including paclitaxel. Therefore, TP **5** was selected to further evaluation based on its high antitumor activity.

### 2.2. Synthesis of TPs-NPs

First, the solubility of TP **5** was investigated at 25 °C and presented in Table 1. The solubility of TP **5** was only 26.36 μg/mL in deionized water and 27.64 μg/mL in PBS (pH 7.4), which were very limited. The solubility was higher in organic solvents. However, organic solvents are toxic and not suitable as pharmaceutical vehicles. Therefore, HSA-based nanoparticles as drug carrier were employed to overcome these obstacles.

For the synthesis of TP **5**-NPs, we found that the fate of TP **5**-NPs was determined by temperature and concentration of TP **5**. Moreover, the phenomenon that the solubility of TP **5** in ethanol increases with the temperature increase was also observed. Nevertheless, it was doubtful that the structure of TP **5** will be damaged as the temperature increases. The stability of TP **5** was investigated at different temperature by HPLC and the results are shown in Table 2. The detected concentrations were close to calculated values when the temperature increases from 25.0 to 60.0 °C. In addition, RSD values were all less than 3.0, which illustrate that the structure of TP **5** was stable when the temperature was increased to 60 °C [19]. Due to using HSA-based nanoparticles as a drug carrier, the structure of HSA will be damaged when the temperature was higher than 60 °C [20]. Therefore, the temperature was not increased during the preparation of TP **5**-NPs.

In order to optimize the preparative parameters of TP **5**-NPs, temperature and concentration of TP **5** were discussed. Table 3 gives particle size, PDI, and ξ potential of obtained TP **5**-NPs under different conditions. As shown in Table 3, TP **5**-NPs can be successfully prepared. When the temperature was 25.0 °C, the average size of TP **5**-NPs was ~425.9 nm and PDI was ~0.748. Increasing temperature to 40.0 °C, particle size gradually decreases with the concentration of TP **5** increasing to 4.0 mg/mL. However, due to the limits on the solubility of TP **5**, the particle size reaches 1084.0 nm and PDI was 0.817. Continuously increasing to 60.0 °C, the average size decreases from ~247.5 to 205.4 nm, PDI was all less than 0.200, which indicate that the dispersibility of TP **5**-NPs was benign. Unfortunately, the particle size and PDI of prepared TP **5**-NPs with a concentration of 7.0 mg/mL was unpromising. Meanwhile, ξ potential of prepared TP **5**-NPs all possesses negative zeta and have no visible difference. Both zeta potential and particle size play significant roles in a nano-delivery system. Nano-delivery systems with a size not exceeding 200 nm can be easily accumulated in the tumor by EPR effect [21,22]. In addition, TP **5**-NPs with negative zeta potential can minimize nonspecific binding to endothelium [23]. Therefore, the optimum concentration for preparing TP **5**-NPs was 6.0 mg/mL and at 60.0 °C.

For TP **5**-NPs synthesized under best optimum conditions, the drug-loading capacity (EC), drug encapsulated efficiency (EE) and solubility of TP **5** were measured by HPLC. TP **5**-NPs possessed 99.59% of EE, 3.70% of EC and 691.4 μg/mL of solubility in HSA aqueous solution. The solubility of TP **5** was increased by 26.2 times.

### 2.3. The Characterization of TP 5-NPs

Figure 2 presents the TEM image, particle size, ξ potential and stability of obtained TP **5**-NPs. TP **5**-NPs have a nearly spherical shape and uniform size with a diameter of about 170 nm (Figure 2A). In addition, it can be seen from Figure 2B that the particle size of TP **5**-NPs was 205.4 nm and ξ potential was -23.2 mV. Meanwhile, stability result indicates that the particle size of TP **5**-NPs have no apparent variations within 72 h and have a good stability (Figure 2C), suggesting meet the requirement of drug delivery system.

### 2.4. Cellular Uptake In Vitro

The cellular uptake of TP **5**-NPs is prerequisite for their antitumor effects. Flow cytometry and laser confocal scanning microscopy (LCSM) were adopted to investigate the cellular uptake of TP **5**-NPs in vitro. As shown in Figure 3, uptake of FITC-labeled NPs (FITC-TP **5**-NPs) by two cell lines showed a time-dependent manner. Compared to untreated cells, a clear shift was observed. Cellular uptake increased by 86.5% in HepG2 and 96.5% in SMMC-7721 cells (Figure 3C), implying that FITC-TP **5**-NPs indeed enter into cells and the process was time-dependent.

Figure 4 is CLSM images of HepG2 and SMMC-7721 cells after incubation with FITC-TP **5**-NPs for 1, 2 and 4 h. The images indicated that FITC-TP **5**-NPs were localized within the cytoplasm and nuclei of cells. The behavior further suggests that FITC-TP **5**-NPs were indeed uptake by cells and would be beneficial for tumor therapy.

### 2.5. Drug Release Kinetics of TP 5-NPs

Figure 5 shows the release profiles of free TP **5** and TP **5**-NPs in vitro. Free TP **5** suspension was released ~20% due to low solubility in PBS over 120 h. TP **5** showed quick release from TP **5**-NPs relative to free TP **5** suspension and 75.0% was released over 120 h. This proved that TP **5** was covered in nanoparticles and could maintain sustained release.

### 2.6. Cytotoxicity of TP 5-NPs In Vitro

HepG2 and SMMC-7721 cell lines were employed to evaluate the cytotoxicity of TP **5**-NPs in vitro by MTT assay. Different concentrations of TP **5** or TP **5**-NPs were used to test cell viability at incubating time of 24, 48 and 72 h, the results are given in Figure 6. Both free TP **5** and TP **5**-NPs exhibited dose and time-dependent cytotoxicity. The cells were completely killed when the concentration of free TP **5** exceeds 40.0 μg/mL. Due to the sustained release of TP **5** from TP **5**-NPs, the cytotoxicity of TP **5**-NPs was initially lower than TP **5** alone. However, the difference disappeared in later time points. These results suggested that TP **5** can be efficiently released from nanoparticles and NPs delivering TP **5** is a promising strategy to treat cancer.

### 2.7. Mitochondrial Transmembrane Potential Assay

In order to prove TP **5** could induce tumor cell apoptosis, ΔΨ_m_ transformation for HepG2 and SMMC-7721 cells was explored by JC-1 dyeing [24,25]. In general, JC-1 dye accumulates in healthy mitochondrial to form a polymer and give off a strong red fluorescence. While in unhealthy mitochondrial, JC-1 dye exists in the cytoplasm and produce green fluorescence [26]. Therefore, the change in color will reflect variation in ΔΨ_m_ and red/green fluorescence ration indicates mitochondrial depolarization.

Figure 7 presents ΔΨ_m_ changes in HepG2 and SMMC-7721 cells treated with different formulations. The merged images treated with free TP **5** or TP **5**-NPs showed that the majority of cells presented strong green fluorescence. Furthermore, as shown in Figure 7B, the red/green fluorescence ratio of TP **5** and TP **5**-NPs were all low, suggesting the plentiful apoptotic activity of TP **5**.

### 2.8. ROS Assay

Mitochondria play an important role in the process of cell proliferation and metabolism. High-level ROS will result in mitochondria dysfunction [19,26]. Figure 8 shows the ROS level of two cell lines (HepG2 and SMMC-7721 cells) treated with free TP **5** or TP **5**-NPs, they all induce ROS generation in two cell lines. Free TP **5** and TP **5**-NPs treated cells present bright green fluorescence. Therefore, the results in Figure 8B indicated that the levels of ROS induced by free TP **5** and TP **5**-NPs were higher than those of control cells, which suggested a better anti-tumor effect.

## 3. Discussion

Recently, TPs have attracted considerable attention due to their high antitumor activity [3,4]. But low solubility and nonspecific limit its application or result in its poor curative effect, high toxic for healthy tissues or side-effect including lesions in the gastrointestinal tract, hair loss and nausea [7].

In this work, the cytotoxicity of 6 TPs was evaluated by MTT assay. According to the IC_50_ of paclitaxel (35.92 μg/mL for HepG2 and 35.33 μg/mL SMMC-7721at 24 h), 30.0 μg/mL of TPs was employed to evaluate their cytotoxicity. By comparison, TP **5** presents high antitumor activity. In order to improve its solubility and tumor bioavailability, HSA, as a drug carrier, plays a significant role in TP **5**-loaded self-assembling nanoparticle, which is expected to result in fewer side effects for health tissue than free TP **5**.

For the synthesis of TP **5**-NPs, the fate of TP **5**-NPs was determined by temperature and the concentration of TP **5**. Therefore, temperature and the concentration of TP **5** were discussed in this work. It can be seen from Table 3 that the particle size was large and the dispersibility was unsatisfactory when the temperature was 25 °C and the concentration of TP **5** was 2.0 mg/mL, which may be attributed to the limitation of solubility. Higher temperature contributes to the unfolding of protein chains and the hydrophobic groups were exposed. Besides, higher temperature results in the larger size of nanoparticles and a larger amount of TP **5** covalent linked with HSA [1]. Therefore, the particle size of TP **5**-NPs prepared at 60 °C was all larger than prepared at 40 °C. However, increasing concentration of TP **5** induced reduction of the mean diameter of TP **5**-NPs, which may be attributed to stronger force between TP **5** and NPs. The concentration of TP **5** was increased and the structure of TP **5**-NPs become more integrated [27]. In addition, due to the low solubility of TP **5**, TP **5**-NPs obtained at a concentration of 5.0 mg/mL at 40 °C and 7.0 mg/mL at 60 °C were all unsatisfactory. The average particle size of TP **5**-NPs under the recommended optimum conditions was 205.4±3.44 nm. TP **5**-NPs within this size can be easily accumulated in tumor tissue by EPR effect and avoid non-specific systemic elimination by reticuloendothelial cells and phagocytes [22,23]. Moreover, TP **5**-NPs can diminish clearance and realize long circulation in blood. In addition, TP **5**-NPs have negative zeta potential, which can minimize nonspecific binding to endothelium.

It was noticeable that the particle size was ~170 nm by TEM, smaller than the average size determined by DLS (205.4 ± 3.44 nm) at 25 °C. This was because DLS showed HSA nanoparticles in the hydrated state, while TEM displayed particle size in the dried state [28,29]. A favorable stability of TP **5**-NPs was ensured in the study of stability for 72 h in vitro, which can prolong retention of TP **5**-NPs in tumor tissue [30]. For drug release assay in vitro, the release of free TP **5** was slower than TP **5**-NPs. This was because TP **5** was dispersed in PBS in the form of a suspension. Only a very small portion of the drug was solubilized due to its low solubility of 27.64 μg/mL. TP **5** solubility in the acceptor medium, which contained 20% ethanol, was 88.15 μg/mL, which was still quite low. During dialysis, the TP **5** suspension is equilibrated with the acceptor medium relatively quickly. However, the slow rate of dissolution of the TP **5** suspension limits the overall rate of drug release. In contrast, TP **5**-NPs had a much larger surface area compared to the TP **5** suspension, as a result of much smaller particle sizes. Therefore, there was a greatly increased dissolution rate for the drug in the NPs formulation. However, TP **5**-NPs show a sustained release and the initial burst release was observed, which results from TP **5** encapsulated in the outer layers of HSA.

Due to sustained release, the cytotoxic of TP **5**-NPs was lower than TP **5**. However, HSA-based nanoparticles have advantages of being non-toxic, non-immunogenic, biodegradable, biocompatible and promotes tumor delivery in vivo. Flow cytometry and LCSM were applied to observe cellular uptake of FITC-TP **5**-NPs in vitro. Fluorescence intensity increased over time. A difference of fluorescence intensity in two cell lines was found, which may be attributed to the differences in their biological behavior and genetic background [1]. As shown in the result of cellular uptake, we inferred that TP **5**-NPs could enter into tumor cells by cellular uptake and delivery of TP **5**. Moreover, low cytotoxicity of the drug carrier [1], excellent stability and sustained release of TP **5**-NPs indicate the potential of such formulation for a meaningful antitumor drug delivery.

## 4. Materials and Methods

### 4.1. Materials

Human serum albumin (HSA, 200 mg/mL) was obtained from Octapharma Pharmaceutical Production Co., Ltd. (Vienna, Austria). Sodium chloride, potassium chloride, potassium dihydrogen phosphate, dipotassium hydrogen phosphate were all from Sinopharm Chemical Reagent co. Ltd. (Shanghai, China). Fluorescein isothiocyanate (FITC), 4′,6-diamidino-2-phenylindole (DAPI) and 3-(4, 5-dimethylthiazol-2-yl)-2,5-diphenyltetrazolium bromide (MTT) were all from Sigma-Aldrich (Saint Louis, MO, USA). Fetal bovine serum (FBS), high glucose Dulbecco’s Modified Eagle’s Medium (DMEM), streptomycin and penicillin were purchased from Gibco (Gibco BRL Co. Ltd., Gaisburg, MD, USA). HepG2 and SMMC-7721 cell lines were all obtained from American Type Culture Collection (ATCC, Rockefeller, MD, USA). All reagents were used without further purification. Distilled water was used in all experiments.

### 4.2. Methods

#### 4.2.1. Cell Culture and MTT Screening Assay

HepG2 and SMMC-7721 cell lines were cultured and grown in complete DMEM containing 10% FBS and 1% streptomycin and penicillin in 5% CO_2_ at 37 °C. TPs were fabricated using the method as described previously [3]. The cytotoxicity of TPs in HepG2 and SMMC-7721 cell in vitro was evaluated by MTT assay. Firstly, cells with a density of 8 × 10^4^ cells per wells were placed and cultured in 96-well plates and incubated 24 h at 37 °C. Then, 100 μL of TPs DMEM solution (30 μg/mL) was added into each well. After incubated 24 h, 10 μL of MTT solution (5.0 mg/mL) was added and incubated 4 h. Finally, the medium was removed, 100 μL of DMSO was added and OD values were tested using microplate reader (Synergy4, multi-mode microplate reader, BioTek, Winooski, VT, USA) at 490 nm when the formazan precipitate was completely dissolved. It is worth noting that wells containing normal medium without TPs were regarded as controls. In addition, wells which add the medium with paclitaxel were regarded as a positive control.

#### 4.2.2. Synthesis of TP 5-NPs

In order to investigate the solubility and stability of TP **5**, a certain amount of TP **5** was dissolved in several solvents (deionized water, PBS buffered solution pH 7.4, ethanol, ethyl acetate, and dichloromethane) and make it saturate. The solubility was measured by HPLC. In addition, for the stability of TP **5**, 200.0 μg/mL of TP **5** was placed in different temperature (25.0, 40.0 and 60.0) and the concentration was detected and compared with practical concentration.

TP **5**-NPs were synthesized according to the previously reported method [31]. The detailed process was as follows: First, TP **5** was dissolved in 2.0 mL of anhydrous alcohol, and 1.0 mL of sodium chloride solution with a mass fraction of 12.5% and 1.0 mL of HSA were added into the above solution. Secondly, they were mixed and rapidly poured into 110.0 mL of 65.0 °C deionized water under rapid stirring. Then, the solution was immediately cooled to 4.0–8.0 °C. The obtained TP **5**-NPs were concentrated by hollow fiber column with a molecular weight cutoff of 50.0 KD to remove unencapsulated TP **5**. Finally, TP **5**-NPs was freeze-dried and stored at 4.0 °C. In order to optimize preparative parameters of TP **5**-NPs, temperature and the concentration of TP **5** were discussed, which were given in Table 4. The particle size, dispersibility, and stability were investigated to determine the optimum parameters for preparing TP **5**-NPs.

#### 4.2.3. Synthesis of FITC-Labelled NPs

FITC-labeled NPs were obtained according to the previously reported method [32]. Firstly, 5.0 mg/mL of HSA was obtained by diluting using a Na_2_CO_3_/NaHCO_3_ buffer solution with a concentration of 25.0 mM and pH 9.8. Then, FITC was added and make a final concentration of 0.1 mg/mL and stirred overnight at 4.0 °C. Finally, FITC-labelled HSA was obtained by centrifuged and washed using ultrafiltration tubes (molecular weight cutoff of 50.0 kD) at 5000 rpm for 10.0 min. FITC-NPs were prepared by the same procedure as TP **5**-NPs.

#### 4.2.4. The Characterization of TP 5-NPs

A series of characterizations were applied to investigate the structures of TP **5**-NPs. Firstly, dynamic light scattering (DLS, Nano-ZS ZEN3600, Malvern, UK) was used to measure particle size, PDI and ξ potential of TP **5**-NPs. TEM (JEOL JEM 2100, Tokyo, Japan) was applied to observe microstructure of TP **5**-NPs.

#### 4.2.5. Cellular Uptake of TP 5-NPs In Vitro

Cellular uptake of TP **5**-NPs in vitro was quantitatively analyzed by flow cytometry (Coulter Epics XL, Beckman Coulter, Kraemer Boulevard Brea, CA, USA). First, HepG2 and SMMC-7721 cell (2 × 10^5^ cells per well) were cultured in 6-well plate for 24 h. Secondly, FITC-TP **5**-NPs were added and incubated for 2, 4 and 6 h. The cells were trypsinized and washed 3 times by PBS (pH = 7.4). After centrifugation at 1000 rpm for 5 min, cells were fixed with 400 μL of 4% formaldehyde solutions.

Cellular uptake of TP **5**-NPs in vitro was qualitatively analyzed by LCSM (Carl Zeiss; Jena, Germany). Two cell lines with 2 × 10^5^ cells per well were cultured in a confocal dish for 24 h, After that, FITC-TP **5**-NPs were added and cultured for another 1, 2 and 4 h. The medium was removed and washed for 3 times by PBS. Then, 500 μL of 4% formaldehyde solution were added and immobilized for 15 min. Remove solution, continuously wash for 3 times by PBS. 350 μL of 2.0 μg/mL of 4′,6-diamidino-2-phenylindole (DAPI) was applied to stain the cell nucleus for 5 min. Finally, continuously wash for 3 times by PBS.

#### 4.2.6. Drug Loading and Release of TP 5-NPs

The EC, EE, and solubility of TP **5**-NPs were measured using High-Performance Liquid Chromatography (HPLC, LC-20AD, Shimadzu Corporation, Kyoto, Japan) with an HC-C18 column (5 μm, 4.0 × 250 mm). The mobile phase was methanol/water (90:10), the flow rate was set up as 1.0 mL/min and 30 °C of column temperature was kept. TP **5**-NPs were demulsified by acetonitrile. Briefly, TP **5**-NPs were diluted in a five-fold volume of acetonitrile and sonicated for 15.0 min. After that, the suspension was centrifuged at 12,000 rpm for 5.0 min, the concentration of the supernatant was detected. EC and EE can be calculated by the following equations:(1)$$\mathrm{EC}\;=\;\frac{m(\mathrm{TPs})}{m(\mathrm{TPs}-\mathrm{NPs})}\times 100\%$$
(2)$$\mathrm{EE}=\frac{m(\mathrm{TPs})}{m(\mathrm{TPs})+m(\mathrm{free}\;\mathrm{Tps})}\times 100\%$$
where *m*(TPs) stands for the mass of TPs in TP-NPs (mg), *m*(TPs−NPs) and *m*(free TPs) present the mass of TP-NPs (mg) and of TPs in the filtrate (mg), respectively.

The release of TP **5** from TP **5**-NPs in vitro was determined in phosphate buffer saline (PBS, pH 7.4) containing 20.0% *v*/*v* ethanol (the solubility of free TP **5** in the release medium was 88.15 μg/mL) by dialysis method [31,32,33]. The detailed process was as follows: first, TP **5** and TP **5**-NPs were all dissolved or dispersed in 1.0 mL PBS (pH 7.4), respectively. Then, they were placed in a dialysis bag (the molecule weight cutoff was 12.0 kD). After that, they were placed in 50.0 mL release medium PBS containing 20.0% *v*/*v* ethanol and maintained at 37 °C with continuous stirring at 150 rpm. Sink conditions were maintained. At predetermined time intervals, 1.0 mL release medium was withdrawn and replaced with 1.0 mL 37 °C fresh release medium. Finally, the released capacity of TP **5** was detected by HPLC.

#### 4.2.7. Cytotoxicity of TP 5-NPs In Vitro

The cytotoxicity of TP **5**-NPs in two cell lines in vitro was assessed. Briefly, cells with a density of 8 × 10^4^ cells per wells were placed and cultured in 96-well plates. After incubated 24 h at 37 °C, 100 μL of medium containing different concentrations of TP **5**-NPs or TP **5** (10.0, 20.0, 30.0, 40.0 and 50.0 μg/mL) was added to each well and cultured another 24, 48 or 72 h at 37 °C (wells containing the normal medium without formulation were regarded as controls). After that, 10.0 μL of MTT (5.0 mg/mL) was added into each well and incubated for 4.0 h. Finally, the medium was removed and 100.0 μL of DMSO was added. OD values were tested by microplate reader at 490 nm when the formazan precipitate was completely dissolved.

#### 4.2.8. Mitochondrial Transmembrane Potential Assay

In general, cell apoptosis is often accompanied by the transformation of mitochondrial transmembrane potential (ΔΨ_m_) [24]. JC-1 dye as a fluorescent probe widely used to detect ΔΨ_m_ which possesses potential-dependent accumulation in the mitochondrial [24,25]. In order to verify whether TP **5**-NPs can induce cancer cell apoptosis, ΔΨ_m_ was measured. Firstly, HepG2 and SMMC-7721 cell with a density of 2 × 10^5^ cells per wells in 1.0 mL of medium all were seeded in a 12-well plate and incubated for 24 h. After that, 30.0 μg/mL of TP **5**-NPs or TP **5** was added and cultured for another 12 h, in which wells containing normal medium without drugs were regarded as controls. Cells were washed using PBS for three times, incubated for 10 min in 10.0 μg/mL of JC-1 dye in the dark. The level of mitochondrial depolarization was observed by inverted fluorescent microscope and quantitatively analyzed by imaging software Image J [26].

#### 4.2.9. The Generation Level of Reactive Oxygen Species

The experimental procedure was similar to the part 4.2.8. Similarly, HepG2 and SMMC-7721 cell were seeded in a 12-well plate and incubated for 24 h. 30.0 μg/mL of TP **5**-NPs or TP **5** was added and cultured for another 12 h. After that, 10 μM 2′, 7′-dichlorodihydrofluorescein diacetate (DCFH-DA, Sigma-Aldrich, Saint Louis, MO, USA) was added and incubated for 10.0 min in the dark [26,34,35]. Inversed fluorescent microscope was applied to observe fluorescence images [26].

#### 4.2.10. Statistical Analysis

Student’s t-test was applied to analyze experimental data. *p* < 0.05 stands for significant difference while *p* < 0.01 indicates a highly significant difference. Mean ± standard error (SD) was used to present the experimental results.

## 5. Conclusions

In this work, six TPs were synthesized and their biological activities were evaluated. TP **5** was identified as a potential anticancer agent based on its ability to inhibit tumor cell growth. In addition, the solubility and tumor bioavailability of TPs was improved by loading into self-assembling HSA nanoparticles. We believe that the data obtained support further development of TP **5**-NPs as a potential anticancer chemotherapeutic. In future work, we plan to evaluate the pharmacodynamics and pharmacokinetics of TP **5**-NPs in vivo.

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
