# Peer review of "Thiophene Derivatives as Anticancer Agents and Their Delivery to Tumor Cells Using Albumin Nanoparticles"

_molecules, 2019, doi:10.3390/molecules24010192_

Reviewer 1 Report

The experimental design is interesting, however some claims are misinterpreted and some data is conflicting, not supporting the conclusions. Please find my comments below:

1.       The English and style should be thoroughly revised. Authors use different tenses in the same section sometimes.

2.       The methods description for drug release is confusing and additional information should be included (solubility of the drug in the release medium, if sink conditions were maintained).

3.       This reviewer suggests that authors should include more information in the legends of the figures. Figures and legend should be self-explanatory without the need to search for information within the text.

4.       Why is the release of TP 5 in the free form so much slower than the TP 5-NPs? Was a different release medium used? Were sink conditions maintained? In the free form, the only barrier for drug release is the diffusion of the drug from one side to the other of the dialysis membrane, whereas for NPs, there are roughly 2 stages: release of drug from NPs and then diffusion through the membrane. Hence, when in NPs, the drug release profile would be expected to be slower or at least more similar to the free drug.

5.       Still associated to the comment above, authors claim in the discussion “Due to sustained release, the cytotoxic of TP 5-NPs was lower than TP 5.” This assumption (despite being reasonable) seems to contradict with the assumptions for the in vitro release data.

6.       In the discussion, the authors say “Moreover, low cytotoxicity, excellent stability and sustained release of TP 5-NPs indicate the potential of such formulation for a meaningful antitumor drug delivery.” For a meaningful antitumor drug delivery, is low cytotoxicity a desirable feature?

7.       The statement regarding paclitaxel in the discussion should be removed since no cytotoxicity data is provided for paclitaxel.

8.       Statements about △ Ψm and ROS levels in the results section contradict conclusions drawn in the discussion section.

Author Response

Response to Reviewer 1 Comments:

The experimental design is interesting, however some claims are misinterpreted and some data is conflicting, not supporting the conclusions. Please find my comments below:

Response: We appreciate these valuable comments. We have made extensive changes in our revised manuscript.

1. The English and style should be thoroughly revised. Authors use different tenses in the same section sometimes.

Response: The English grammar and style of the manuscript have been checked and revised by a professional. The corresponding revisions have been marked RED in the revised manuscript.

2. The methods’ description for drug release is confusing and additional information should be included (solubility of the drug in the release medium, if sink conditions were maintained).

Response: The description for drug release has been revised. The solubility of the drug in release medium and sink conditions have been added in the revised manuscript. The release medium was PBS containing 20.0% v/v ethanol. The solubility of free TP5 in the release medium was 88.15 μg/mL. Sink conditions were maintained.

3. This reviewer suggests that authors should include more information in the legends of the figures. Figures and legend should be self-explanatory without the need to search for information within the text.

Response: Corresponding information in the legends of the figures has been supplemented and marked RED in the revised manuscript.

4. Why is the release of TP 5 in the free form so much slower than the TP 5-NPs? Was a different release medium used? Were sink conditions maintained? In the free form, the only barrier for drug release is the diffusion of the drug from one side to the other of the dialysis membrane, whereas for NPs, there are roughly 2 stages: release of drug from NPs and then diffusion through the membrane. Hence, when in NPs, the drug release profile would be expected to be slower or at least more similar to the free drug.

Response: The release of free TP 5 was slower than TP 5-NPs. This was because the solubility of TP 5 was very low in PBS (pH=7.4). The solubility of TP 5 was only 27.64 μg/mL in PBS and TP 5 was dispersed in PBS in the form of suspension, which made it difficult to be released from dialysis membrane. However, TP 5-NPs were completely dispersed in PBS and TP 5 was able to be released from NPs and then diffusion through the membrane. Therefore, the release of TP 5 in the free form was slower.

In this manuscript, the release experiments were performed using the same release medium and sink conditions were maintained.

5. Still associated with the comment above, authors claim in the discussion “Due to sustained release, the cytotoxic of TP 5-NPs was lower than TP 5.” This assumption (despite being reasonable) seems to contradict with the assumptions for the in vitro release data.

Response: Cytotoxicity assay was performed under different conditions from the drug release studies and cell death occurred at lower concentrations, which explains the discrepancy.

6. In the discussion, the authors say “Moreover, low cytotoxicity, excellent stability and sustained release of TP 5-NPs indicate the potential of such formulation for a meaningful antitumor drug delivery.” For a meaningful antitumor drug delivery, is low cytotoxicity a desirable feature?

Response: We appreciate this valuable comment. This “low cytotoxicity” refers to drug carrier show low cytotoxicity to healthy tissues. The cytotoxicity of drug carrier was evaluated in our previous work [1]. The corresponding expression has been revised in the revised manuscript.

7. The statement regarding paclitaxel in the discussion should be removed since no cytotoxicity data is provided for paclitaxel.

Response: Thank you for your reminder and the statement regarding paclitaxel in the discussion has been removed.

8. Statements about △Ψm and ROS levels in the results section contradict conclusions drawn in the discussion section.

Response: We appreciate this valuable comment. Corresponding statements have been revised.

Reference:

1.      Qu, N.; Sun, Y.T.; Xie, J.; Teng, L.S. Preparation and evaluation of in vitro self-assembling HSA nanoparticles for cabazitaxel. Anti-Cancer Agent in Me 2017, 17, 294-300, DOI: 10.2174/1871520616666 160526103102.

Reviewer 2 Report

-References are not in numerical order.

-"Both zeta potential and particle size play a significant role in nanocarrier.": please edit the sentence.

-"For two cell lines, free TP 5 and TP 5-NPs all exhibited dose-dependent and time-dependent." maybe is "Both free TP 5 and TP 5-NPs exhibited a dose/time-dependent trends in the employed cell lines".

-Section 2.6: I suggest to clearly state that despite the less cytotoxicity of TP5-HSA respect TP5 alone, the inclusion of the drug in a carrier would potentially improve its action on neoplasms in a complex environment while reducing side effects.

Author Response

Response to Reviewer 2 Comments

1.References are not in numerical order.

Response: The numerical order of references is checked and revised in our revised.

2. "Both zeta potential and particle size play a significant role in nanocarrier." please edit the sentence.

Response: The sentence has been revised.

3. "For two cell lines, free TP 5 and TP 5-NPs all exhibited dose-dependent and time-dependent." maybe is "Both free TP 5 and TP 5-NPs exhibited a dose/time-dependent trends in the employed cell lines".

Response: Thank you for your valuable comment. The corresponding sentence has been revised.

4. Section 2.6: I suggest to clearly state that despite the less cytotoxicity of TP5-HSA respect TP5 alone, the inclusion of the drug in a carrier would potentially improve its action on neoplasms in a complex environment while reducing side effects.

Response: We appreciate this valuable comment. We have revised the discussions accordingly.

Reviewer 3 Report

The manuscript by Guangsheng Cai et al. describes screening of a potent thiophene derivative and establishment of a delivery system using an albumin nanoparticle with some tuning. A screened lead was tailored with an albumin nanoparticle to get over poor solubility and high toxicity.  But, the targeting motif on some tumor cells does not look like very clear. The design of TP-NPs to aim on cancer cells might introduce any ligand on NP to lead gp60 trancytosis. This albumin-based NP might result in unspecific delivery.

Author Response

Response to Reviewer 3 Comments:

The manuscript by Guangsheng Cai et al. describes screening of a potent thiophene derivative and establishment of a delivery system using an albumin nanoparticle with some tuning. A screened lead was tailored with an albumin nanoparticle to get over poor solubility and high toxicity. But, the targeting motif on some tumor cells does not look very clear. The design of TP-NPs to aim at cancer cells might introduce any ligand on NP to lead gp60 transcytosis. This albumin-based NP might result in unspecific delivery.

Response: Thank you for your valuable comment. We agree albumin-based NP might result in unspecific delivery. In future work, we plan to introduce some ligand on NPs to promote gp60 transcytosis.

Round  2

Reviewer 1 Report

Dear authors,

the point made in question 4 still intrigues this reviewer. If TP 5 was dispersed in PBS in the form of suspension, it means it is not soluble in the donor media. This fact would prevent the drug from crossing the dialysis membrane, affecting the release pattern for the free drug. It would be imperative that the drug is soluble so it can cross the membrane and if only a very small part of it is solubilized, then I agree with the authors, it will delay the release rate of the free drug. On the other hand, if the acceptor medium contains ethanol, which is a very small molecule, it will diffuse towards inside the dialysis membrane until it reaches equilibrium, which should not take long time. 
While I still believe there are flaws in the drug release assay, the authors provided their explanation for those results. They should clear the confusions surrounding the study, mostly for themselves.

Author Response

Responses to Reviewer 1 Comments

The point made in question 4 still intrigues this reviewer. If TP 5 was dispersed in PBS in the form of suspension, it means it is not soluble in the donor media. This fact would prevent the drug from crossing the dialysis membrane, affecting the release pattern for the free drug. It would be imperative that the drug is soluble so it can cross the membrane and if only a very small part of it is solubilized, then I agree with the authors, it will delay the release rate of the free drug. On the other hand, if the acceptor medium contains ethanol, which is a very small molecule, it will diffuse towards inside the dialysis membrane until it reaches equilibrium, which should not take long time. 

While I still believe there are flaws in the drug release assay, the authors provided their explanation for those results. They should clear the confusions surrounding the study, mostly for themselves.

Response: We appreciate this valuable comment and would like to provide further clarification. For the release study, TP 5 was dispersed in PBS in the form of a suspension.  Only a very small portion of the drug was solubilized due to its low solubility of 27.64 μg/mL. TP5 solubility in the acceptor medium, which contained 20% ethanol, was 88.15 μg/mL, which was still quite low. During dialysis, the TP5 suspension is equilibrated with the acceptor medium relatively quickly.  However, the slow rate of dissolution of the TP5 suspension limits the overall rate of drug release. In contrast, TP 5-NPs had a much larger surface area compared to the TP5 suspension, as a result of much smaller particle sizes. Therefore, there was a greatly increased dissolution rate for the drug in the NPs formulation. These discussions have been added to the revised manuscript. 

This manuscript is a resubmission of an earlier submission. The following is a list of the peer review reports and author responses from that submission.

Round  1

Reviewer 1 Report

The manuscript by Guangsheng Cai et al. describes formation of HSA-based nano particle to render functional, but poor soluble anti-cancer chemical agent effective under aqueous condition. Most likely, the work presented here is technically sound and generally well presented as enough informative to relevant readers. 

For exploration of Thiopen derivatives, MTT assay was achieved and one of them was chosen as a potential drug lead. After this data, docking study might be useful to explain drug activity or solubility issue. The reason why HSA-nano encapsulation is needed in this study should be addressed with experimental data before TP-5 NP formulation.But, this Thiopen derivative was encapsulated into HSA-based nanoparticle without checking any solubility issue. Before this paper, I would recommend this author to try any research  about  Thiopen-based anti-cancer agent exploration before this paper. Sometimes, drug effectiveness might come not only from fitting onto the target, but also from solubility.

Reviewer 2 Report

In thismanuscript, the Authors report the synthesis of thiophene-loaded albumin nanoparticles and assess their product in a couple of cell lines. 

The manuscript can be interesting but the conclusions are not supported by data. Moreover, the manuscript needs both a complete restyling and English revision. I cannot support its publication in Molecules at this time.

-      the English of the manuscript must be improved. The text has to be restyled. For example, section 2.8 should be reported after section 2.3. Also section 2.1 should be reported after section 2.3 and before section 2.8.

-      Section 2.4: how have the Authors evaluate the release of TPs 5 free drug? Release from what?

-      It is not reported nor discussed the synthesis and characterizations of the six TPs.

-      In section 2.1 and Fig.1 is reported the MTT assay on the six TPs. Why TPs 5 works “better”? Maybe it is more soluble? How is the concentration of the drugs employed?

-      In the text Authors state about EPR. It is now well accepted that EPR is a complex phenomenon (for example refer to doi: 10.1016/j.addr.2013.11.009). Thus, Authors should take care while stating about EPR.

-      In the introduction the Authors should add a sentence in which they discuss the presence of a number of nanoplatforms to increase the drug delivery, referring for example to doi: 10.1021/acs.bioconjchem.7b00664 and 10.1038/sj.bjc.6604483)

-      Authors state that drug delivery systems have to enhance the bioavailability of the drugs to tumors and report the example of abraxane. Thus, a drug that is able to kill cancer cells only at a concentration that can be toxic for the rest of the organism is implemented in the action due to the conjugation to a nanoplatform. In this manuscript, the drug employed by Authors (TPs 5) “works” better alone than if conjugated in HSA nanoparticles. In this case seems unnecessary to load the drug in a nanoplatform. Have the Authors employed the same concentration of drugs in their cytotoxicity investigations? If yes, the conclusion “And the solubility and bioavailability of TPs was improved by loading into nanoparticles..” is not supported.

Reviewer 3 Report

Dear Authors,

please find below minor concerns about the manuscript:

1.       A thorough review in the grammar and structure is suggested. Also, confirm if all the sections that are referred are correct (e.g. line 334 mentioning part 4.1.7, but means 4.2.7).

2.       Regarding the in vitro drug release, is the release medium PBS containing 20% v/v ethanol? This section in the methods is slightly confusing. What is the solubility of free TPs in the release medium? Please mention the solubility and if sink conditions were maintained.

3.       Is there a hypothesis for the anticancer mechanism of TPs? How is it comparable to paclitaxel?

4.       For the mitochondrial transmembrane potential assay, it was analyzed after 12h of incubation with the drugs. The authors mention lower apoptotic activity when incubated with paclitaxel when compared to TPs 5. The main antitumor effect of paclitaxel is related to stabilization and microtubules, which leads to mitotic arrest and only later to apoptosis. This process should take longer than 12h. Could the lower apoptotic events from paclitaxel be attributed to the 12-hour incubation time? Same question would be for the ROS evaluation experiment.

5.       In the in vitro cellular uptake, line 180, the authors stated that fluorescence intensity is stronger at 2h than that at 4h. This is not what Figure 7 shows.

6.       The authors mention the NPs were freeze-dried. It is generally required to confirm if there is any effect from the process in the nanoparticles (aggregation, size increase, drug leakage, increase in PDI, etc.). Did the authors evaluate the effect of freeze-drying in the characteristics of the NPs? Did they use any cryoprotectant?

7.       The confocal studies do not include optical z-stacking so authors should be careful when making claims regarding the internalization of the formulation.